# Inequalities in Childhood Immunisation in South Asia

**DOI:** 10.3390/ijerph20031755

**Published:** 2023-01-18

**Authors:** Madhu Sudhan Atteraya, In Han Song, Nasser B. Ebrahim, Shreejana Gnawali, Eungi Kim, Thakur Dhakal

**Affiliations:** 1Department of Social Welfare, Keimyung University, Daegu 42601, Republic of Korea; 2Department of Social Welfare, Yonsei University, Seoul 03722, Republic of Korea; 3Department of Public Health, Keimyung University, Daegu 42601, Republic of Korea; 4International Affairs Team, Keimyung University, Daegu 42601, Republic of Korea; 5Department of Library and Information Science, Keimyung University, Daegu 42601, Republic of Korea; 6Department of Life Science, Yeungnam University, Gyeongsan 38541, Republic of Korea

**Keywords:** inequalities, childhood immunization, South Asia, child health, sustainable development goals

## Abstract

Identifying the inequalities associated with immunisation coverage among children is crucial. We investigated the factors associated with complete immunisation among 12- to 23-month-old children in five South Asian countries: Afghanistan, Bangladesh, India, Nepal, and Pakistan, using nationally representative data sets from the Demographic and Health Survey (DHS). Descriptive statistics, bivariate association, and logistic regression analyses were employed to identify the prevalence and the factors in each country that affect the likelihood of full childhood immunisation coverage. The complete childhood immunisation coverage varied significantly within each country in South Asia. Afghanistan had the lowest immunisation rates (42.6%), whereas Bangladesh ranked the highest in complete childhood immunisation rates, at 88.2%. Similarly, 77.1% of Indian children, 79.2% of Nepali children, and 62.2% of Pakistani children were completely immunised. Household wealth status strongly correlated with full childhood immunisation in Afghanistan, India, and Pakistan at the bivariate level. The results from the logistic regression showed that a higher maternal educational level had a statistically significant association with complete childhood immunisation in all countries compared to mothers who did not attend any school. In conclusion, the study revealed the inequalities of complete childhood immunisation within South Asia. Governments must be proactive in their endeavours to address universal and equitable vaccine coverage in collaboration with national and international stakeholders and in line with the relevant Sustainable Development Goals.

## 1. Introduction

Complete childhood immunisation is central to reaching the Sustainable Development Goals (SDGs) in several areas, including ending poverty and hunger, improving health and quality education, reducing inequalities, and preventing child mortality and morbidity, especially in low- and middle-income countries (LMICs) [1,2,3]. Immunisation is one of the most successful and cost-effective interventions for addressing health and wellbeing challenges in resource-limited countries [4,5]. Consequently, it is crucial to carefully scrutinise the complete vaccine utilisation within countries and beyond. Our study’s samples were from five South Asian countries: Afghanistan, Bangladesh, India, Nepal, and Pakistan. These were used to examine the inequalities in childhood immunisation with the aim of providing insights to the government and international communities that can be used to take appropriate action to ensure child health equity.

South Asia is home to a quarter of the world’s population, and 627 million children live in this region. It is also one of the most densely populated regions, with a highly heterogeneous population in terms of religion, geography, ethnicity, culture, and socioeconomic status. Along with this diversity across and within South Asian countries, disparate health statuses, healthcare policies, services, and delivery systems, including the dissemination and delivery of complete childhood immunisation, have been observed. The evidence [6,7,8] suggests that rapid but inequitable socioeconomic development has accentuated health inequalities, thereby posing an increased number of significant health and welfare challenges, including the burden of infectious and vaccine-preventable diseases. The United Nations International Children’s Emergency Fund (UNICEF) [8] further warns that South Asia could face new health challenges if children are not immunised as sporadic outbreaks of vaccine-preventable diseases (i.e., measles and diphtheria) have emerged in South Asian countries. 

The World Health Organization (WHO) implemented the Extended Programme on Immunization (EPI) in 1974 to ensure that all children from all countries receive complete doses of primary childhood immunisation in order to reduce child mortality worldwide. In 1977, the WHO aimed to immunise every child in the world against diphtheria, pertussis, tetanus, poliomyelitis, measles, and tuberculosis [9]. As per the WHO’s recommendation, a child is considered fully immunised if they receive a dose of the bacillus Calmette-Guerin (BCG), three doses of the diphtheria-tetanus-pertussis (DPT), three doses of the oral polio, and a dose of the measles, mumps, and rubella (MMR) vaccinations [3,10,11]. Nevertheless, over four decades have passed, and complete childhood immunisation remains a grave challenge in most LMICs. 

Reducing child health inequalities, including those in the universal coverage of childhood immunisation, is a global challenge. Even in high-income countries, complete childhood immunisation is a challenge, and vaccine non-utilisation is associated with parental socioeconomic status [12]. Bocquier et al. (2017) [12] revealed that higher socioeconomic status is generally associated with childhood immunisation. However, cognitive factors (i.e., doubts, behavioural intention) mediate the relationship between parental socioeconomic status and childhood vaccination. In high-income countries (e.g., the United Kingdom and Germany), some affluent parents have a stronger sense of ‘manufactured risks’ and ‘distrust toward science’, which causes vaccine hesitancy [12,13]. However, lower socioeconomic status as well as structural and cultural barriers in low-income countries are associated with vaccine non-utilisation. The barriers relevant to vaccine non-utilisation include household poverty and financial deprivation, maternal education, lack of knowledge of vaccines and lack of media exposure, larger family size, place of delivery, antenatal care visits, distance to health facilities, migration, parents’ forgetfulness, distrust of immunisation programmes, time and language barriers, limited human resources, poor infrastructure, and an inadequate supply of vaccines [14,15,16].

Specific to the South Asian context, in Afghanistan, variables such as giving birth in a health facility, a maternal age of 30–39 years, at least four visits to facilities to access antenatal care, visiting health facilities in the past 12 months, paternal professional occupation, being a part of a family that has a more affluent wealth status, and living in the northeast region were positively associated with childhood immunisation status [17,18,19]. Furthermore, the complete childhood immunisation rates for Afghani children were only 37%, which is extremely low, as the national target coverage is 90% [18]. In Bangladesh, India, Nepal, and Pakistan, previous studies reported that demographic factors (i.e., mother’s age, gender of the child), lower socioeconomic status, lack of access to information, lack of access to health care, and lower levels of social development in the region significantly affected childhood immunisation rates [20,21,22,23,24,25,26,27].

Although previous studies have provided insights into the factors that impede or facilitate childhood immunisation in South Asian countries, it is necessary to investigate inequalities in childhood immunisation within and between countries in a single study rather than in a country-specific context. Therefore, we aimed to identify the inequalities that exist in childhood immunisation rates and the sociodemographic factors associated with childhood immunisation in Afghanistan, Bangladesh, India, Nepal, and Pakistan. 

## 2. Materials and Methods

### 2.1. Data

The Demographic and Health Surveys (DHS) program is a worldwide initiative to collect, disseminate, and provide comprehensive data on population health, including childhood immunisation. The survey was conducted by national governments, with technical and financial assistance from the United States Agency for International Development (USAID). The survey was conducted by trained enumerators using a two-stage cluster sampling design, and the data collection procedure follows the standard protocol and instrument, which can be comparable across countries.

We used nationally representative sample sizes from the seventh phase of the five countries’ Demographic and Health Survey (DHS) datasets. The seventh phase is the latest datasets available for further analysis. The data were taken from the 2015 Afghan DHS, the 2017–18 Bangladeshi DHS, the 2019–21 Indian DHS, the 2016 Nepali DHS, and the 2017–18 Pakistani DHS. The DHS has developed the concept of ‘recode files’ across countries and phases based on the unit of analysis (i.e., households, household members, women, children, and so on). We used the Children’s Recode Datasets in which information is provided for every child born in the five years preceding the survey. The DHS interviews mothers and collects information about child health indicators, including prenatal to postnatal care, childhood diseases (e.g., diarrhoea, fever, and cough), the available methods for seeking treatment, and immunisation coverage.

We selected children between 12 and 23 months old to measure whether or not a child received complete vaccination. This is because a child is considered fully immunized if s/he received a BCG vaccine right after birth, three doses each of the DPT and polio vaccines at 6, 10, and 14 weeks of age, and the MMR vaccine at 9 months. We used the data from 5555 Afghan children, 1666 Bangladeshi children, 43,073 Indian children, 1025 Nepali children, and 2314 Pakistani children.

#### 2.1.1. Dependent Variable

The dependent variable, full childhood immunisation, was measured as a child receiving all eight doses of the necessary vaccines as recommended by the WHO’s EPI [10,21,25,28,29]. The immunisation coverage was determined based on the immunisation date and vaccination received written on the report card and the mothers’ reports of vaccination coverage. Those who reported immunisation coverage of all eight doses, including the mothers’ reports on immunisation, were coded as ‘1’ to measure immunisation. Similarly, those who reported ‘No’ were measured as not immunised. After this, we aggregated all values of all eight doses and reassigned the ‘1’ value to children who received all eight doses of the vaccines and were considered fully immunised. Those children who were not immunised or were only partially immunised were grouped as ‘0’ to measure for not fully immunised. 

#### 2.1.2. Independent Variable

We selected socioeconomic and demographic factors as independent variables based on previous studies conducted in South Asian countries [19,20,22,24,25,26,27]. These variables include the mothers’ educational levels, households’ wealth statuses, place of residence, the children’s gender, the number of children, and the mothers’ age at their first birth. 

The DHS provides information on mothers’ educational levels, which are divided into four categories: no education, primary education, secondary education, and higher education. Those who responded as having ‘no education’ were coded as ‘0’, ‘primary education’ was coded as ‘1’, ‘secondary education’ was coded as ‘2’, and ‘higher education and above’ was coded as ‘3’. Similarly, the DHS provides information on household wealth, which is an indicator of a household’s cumulative living standard based on ownership of selected assets, such as televisions and bicycles, materials used for housing construction, and access to water and sanitation facilities. The DHS further classifies this wealth into five categories: poorest, poor, middle, richer, and richest. The wealth index is a categorical variable and was recoded as ‘1’ for ‘poorest’, ‘2’ for ‘poor’, ‘3’ for ‘middle’, ‘4’ for ‘rich’, and ‘5’ for ‘richest’’. The variable, total number of children, was measured as a continuous variable. Mothers below 20 years old were recoded as "0", and mothers above 20 were recoded as 1.

#### 2.1.3. Data Analysis

Firstly, a descriptive statistics analysis was conducted to determine the frequency distribution of the independent and dependent variables. Secondly, chi-square tests were conducted to examine the bivariate association between the categorical dependent and independent variables. We used an independent samples t-test for continuous independent and “grouped” (immunised or not-immunised) dependent variables. Finally, logistic regression was used to identify the factors that affect full immunisation among children. For both bivariate and logistic regression analysis, we used a *p*-value of less than 0.05 to determine the statistical significance. The statistical analyses were performed using the Statistical Package for Social Sciences (V.25) for Windows.

## 3. Results

The full immunisation coverage by each vaccine type is presented in Figure 1, which demonstrates that most children had received the BCG vaccination (Afghanistan: 70%; Bangladesh: 98.2%; India: 94.9%; Nepal: 97.7%; Pakistan: 84%). The vaccination rates for the complete three doses of the polio vaccination were lowest in Afghanistan (59.1%), followed by India (80.6%), Pakistan (83.8%), Nepal (88.6), and Bangladesh (94.7%). The three doses of the DPT vaccination had lower coverage rates among children in Afghanistan (53.2%) and Pakistan (75.1%). Only 55.3% of Afghani children and 68.7% Pakistani children were immunised with the measles vaccine. The figure also shows that 42.6% of Afghan children, 88.2% of Bangladeshi children, 77.1% of Indian children, 79.2% of Nepali children, and 62.2% of Pakistani children were fully immunised. 

Table 1 presents the sample’s descriptive statistics, the bivariate association between the selected independent variables, and the complete vaccination utilisation rates. In Afghanistan, mothers’ educational level, wealth status, places of residence, and the mothers’ age at their first birth had a statistically significant association with complete childhood immunisation. Mothers’ educational level was strongly correlated with full childhood immunization in Bangladesh. In India, mothers’ educational level, wealth, number of children, and the mothers’ age at their first birth showed a significant association with full immunisation. In Nepal, mothers’ educational level, number of children, and mothers’ age at their first birth were significantly associated with complete childhood immunisation. In Pakistan, mothers’ educational levels, wealth statuses, places of residence, and number of children were strongly associated with full childhood immunisation. Overall, mothers’ education was significantly associated with full childhood immunization in all South Asian countries. Household wealth status was associated with full childhood immunization in Afghanistan, India, and Pakistan, but not in Bangladesh and Nepal. 

Table 2 presents the results of the logistic regression analyses of Afghanistan, Bangladesh, India, Nepal, and Pakistan. The results shows that mothers having higher educational levels remained the strongest predictor for the higher likelihood of complete vaccination in all countries (Afghanistan: odds ratio [OR] = 1.85, CI = 1.22–2.81; Bangladesh: OR = 2.93, CI = 1.43–6.00; India: OR = 1.29, CI = 1.18–1.42; Nepal: OR = 3.75, CI = 1.82–7.71; and Pakistan: OR = 4.23, CI = 2.98–6.01) compared to mothers who did not attain any education. Similarly, in terms of wealth status, rich mothers remained a strong predictor for a higher likelihood of childhood vaccination in Afghanistan, India, and Pakistan but not in Bangladesh and Nepal. In Afghanistan, children from richest families were 1.61 times (CI = 1.26–2.04) more likely to have been completely immunised than their poor counterparts. In India, children from the richest families were 1.59 times (CI = 1.44–1.75) more likely to have been completely immunised. In Pakistan, children from the richest families were 1.93 times (CI = 1.30–2.83) more likely to have been completely immunised. The results show that Indian parents living in rural areas were 1.30 times (CI = 1.22–1.39) more likely to immunise their children than those living in urban areas. Female children were less likely to be immunised (OR = 0.95; CI = 0.90–0.99) than male children in India. There were no statistically significant associations between gender and complete childhood immunisation in Afghanistan, Bangladesh, Nepal, and Pakistan. 

Overall, the result depicts that socio-demographic factors were associated with full childhood immunization in South Asia. The higher the mother’s educational level, the more likely they were to use full childhood immunization. Nevertheless, other socio-demographic factors show some discrepancies, e.g., household wealth status did not show any association for full immunisation in Nepal and Bangladesh. Rural children were more likely to use full immunisation only in India but not in other South Asian countries. Only in Afghanistan and India were mothers aged more than 20 years at first birth more likely to immunize their children.

## 4. Discussion

We identified the inequalities in childhood immunisation utilization in the South Asian countries of Afghanistan, Bangladesh, India, Nepal, and Pakistan. Bhutan and Sri Lanka were excluded due to the unavailability of recent datasets from the DHS. Along with exploration of childhood immunisation inequalities among the countries, we also revealed that key socioeconomic and demographic characteristics remain central challenges to the achievement of childhood immunization in South Asia. Therefore, the study’s findings can be helpful in policy development and programme interventions by providing insight into achieving UN SDG 3 for Good Health and Wellbeing in the South Asian context. 

We revealed that South Asian countries struggle to achieve universal childhood immunisation, which directly affects child welfare, including an increase in child morbidity and mortality rates as well as outbreaks of vaccine-preventable diseases. We revealed the inequality of complete childhood immunization, as only 42.6% of Afghan children were fully immunised, while the full vaccination rates in India were only 77.1%, and 62.2% in Pakistan. Bangladesh had the highest full immunisation rates among South Asian countries, at 88.2%, followed by 79.2% in Nepal. These findings provide insights that governments, policymakers, and health-related international organisations can use to take appropriate action in order to ensure complete childhood immunisation in these countries.

World leaders, including the leaders from South Asian countries, committed to the United Nation’s (UN’s) Millennium Declaration and set goals to combat poverty, illiteracy, disease, and gender inequality via the Millennium Development Goals (MDG). MDG 2 aimed to ‘achieve universal primary education’ and ‘ensure that, by 2015, children everywhere, boys and girls alike, will be able to complete a full course of primary schooling’ [30]. Nevertheless, our study revealed that countries were unable to achieve universal primary education. Notably, 85% of women in Afghanistan did not have formal education, followed by 42.7% in Pakistan, 29.5% in Nepal, 22.5% in India, and 6% in Bangladesh. These lower educational levels in these countries have negatively affected children’s quality of life, including childhood immunisation coverage. 

Furthermore, we found that mothers’ educational level was the strongest predictor of childhood vaccination in all countries at the bivariate level. Multivariate logistic regression analysis also showed that higher levels of education in mothers resulted in their being more likely to immunise their children than non-educated mothers. This finding is consistent with several studies conducted in the South Asian context [17,21,25,26,31,32]. Moreover, educational inequalities in childhood immunisation may be more pervasive among specific subgroups of the population based on place of residence, gender, state and substate level, as well as among scheduled castes and tribal populations in India [32] and other South Asian countries.

Household wealth status was a strong predictor of full immunisation at the multivariate level in Afghanistan, India, and Pakistan but not in Bangladesh and Nepal. Previous studies in Bangladesh [33] provided evidence that household wealth status did not have any statistically significant association with BCG vaccination. However, children born to more affluent families were more likely to be immunised with the DPT, polio, and MMR vaccines than those born in the poorest households. Similarly, household wealth status was not a significant predictor of childhood vaccination in Nepal [25]. Both Bangladesh and Nepal adopted the EPI in 1979, and they have consequently improved childhood immunisation rates. In Bangladesh, UNICEF (2021) [34] warns that, although immunisation coverage is high, inequalities persist among children living in urban slums, remote areas of Chittagong Hill Tracts and river island areas, and some tea garden areas. In Nepal, inequalities in complete childhood vaccination have remained a problem among underprivileged castes or ethnic populations and people living in geographically remote regions [25]. Therefore, household wealth status may not significantly predict full immunisation at a national level (macro level) in Bangladesh and Nepal. Other factors, including immediate financial difficulties, living in extreme poverty, and being a child of an underprivileged caste or ethnic household may impede being fully immunised in Bangladesh and Nepal.

We revealed that Indian children living in rural areas were more likely to be fully immunised than their urban counterparts. This finding contradicts the norm that the children residing in rural areas are disadvantaged, especially in LMICs. However, in India, a country with extreme heterogeneity in geography, cultural settings (i.e., North, West, South, Central, East, and Northeast), and between states, the inequalities in these aspects need to be examined [24]. Another challenge in India is increasing urban poverty, as 81 million people (25% of the urban population) live below the poverty line [35]. Thus, childhood immunisation coverage is much more challenging in poor urban areas than in the general urban and rural populations in India [36]. In India, the rural–urban differences in childhood immunisation were possibly reflected due to the impact of India’s Routine Immunisation (RI) programme, which focused on rural areas under the National Rural Health Mission (NRHM). Nevertheless, further study is warranted in order to examine the differences that exist in childhood immunisation in India between the rural and urban populations, including an investigation of the diverse geographical, cultural, and inter-state differences in India. Additionally, the results from the multivariate analysis of India revealed that female children were less likely to be immunised as compared to male children. These gender differences in childhood vaccination in India require further scrutiny to determine why and to what extent female children were less likely to be fully immunised while taking the sociocultural, religious, and geographic heterogeneity of the Indian context into account. 

The study has some limitations. The first limitation is that it used cross-sectional data, thereby creating difficulties in determining a causal relationship; The results only present a snapshot of the data. The second limitation is that the study did not include country-specific sociocultural factors, such as the diversity of castes or ethnicities and religion in these South Asian countries. South Asian countries are highly heterogeneous in terms of socioeconomic aspects, culture, and religion. Further country-specific analysis is needed, including these variables in country-specific contexts. The third limitation of the study is that the study did not analyse the variables related to refrigerators needed for vaccination storage, vaccine stocks, vaccine handling, managing vaccine supply, vaccine providers’ levels of training, or the administration of vaccines. The fourth limitation is that the information on child immunisation was based on immunisation cards and mothers’ self-reports. Such self-reports are subject to social desirability bias in that the mothers are subject to overestimating the information they provide. Finally, categorising the dependent variable into a binary variable (i.e., ‘received full immunisation’ or ‘did not receive full immunisation’) could have led to a loss of information. Nevertheless, the study has strengths as it revealed the inequalities in complete childhood immunisation rates across South Asian countries in a single study. The findings are robust and can be used to design, develop, and implement effective policy and practice interventions in order to enhance equity in childhood immunisation, thereby improving child welfare and health justice for the world’s most vulnerable and underserved children. Furthermore, the DHS also needs to consider the collection of information in national surveys on people’s perceptions regarding vaccine intake (i.e., misinformation, fabrication, distortion, or omission” of scientific facts) that we witnessed during the COVID-19 pandemic [37]. Such information is missing in the DHS survey questionnaire. 

## 5. Conclusions

In conclusion, full childhood immunisation is challenging in low-income countries, including South Asia, due to the complex heterogeneity of culture, geography, and socioeconomic status. A significant gap still exists, thereby delaying the government’s and Global Vaccine Action Plan (GVAP)’ s vision for 2030 to provide full childhood immunisation to every child. We revealed that sociodemographic variables had remained central barrier factors for complete childhood immunisation. Therefore, governments and concerned stakeholders must develop proactive and tailored intervention programmes for universal vaccination coverage addressing social inequalities in line with the UN SDGs, including Goal 1 (No Poverty), Goal 3 (Good Health and Wellbeing), and Goal 4 (Quality Education).

## Figures and Tables

**Figure 1 ijerph-20-01755-f001:**
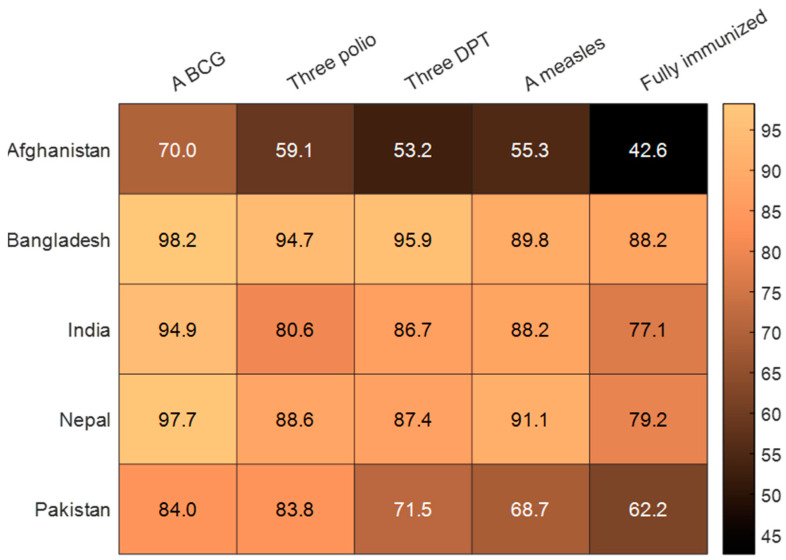
Childhood immunisation by vaccine type in South Asia.

**Table 1 ijerph-20-01755-t001:** Descriptive characteristics of the sample and bivariate association between independent variables and full immunisation.

	Afghanistan		Bangladesh		India		Nepal		Pakistan	
	All population (N; %)	Yes (N; %)	All population (N; %)	Yes (N; %)	All population (N; %)	Yes (N; %)	All population (N; %)	Yes (N; %)	All population (N; %)	Yes (N; %)
Mother’s Education		***		***		***		***		***
None	4698; 84.8	1861; 39.6	100; 6.0	79; 79	8385; 19.5	5833; 69.6	302; 29.5	213; 70.8	1092; 47.2	474; 44.7
Primary	392; 7.0	216; 55.1	470; 28.2	393; 83.6	5124; 11.9	3812; 74.4	207; 20.2	160; 78.4	301; 13.0	217; 72.3
Secondary	364; 6.4	225; 61.8	787; 47.2	711; 90	23,027; 53.4	18,279; 79.4	376; 36.7	310; 82.4	544; 23.5	416; 77.3
Higher	101; 1.8	62; 61.4	309; 18.5	284; 91.9	6537; 15.2	4297; 81.0	140; 13.7	128; 91.4	377; 16.3	306; 82
Wealth		***		*		***				***
Poorest	1001; 18.0	371; 37.1	354; 21.2	302; 85.3	11,255; 26.2	7992; 71.0	254; 24.8	197; 77.6	475; 20.5	189; 40.8
Poorer	1232; 22.2	464; 37.7	344; 20.6	294; 85.5	9970; 23.2	7602; 76.2	230; 24.4	184; 80.0	500; 21.6	278; 57.3
Middle	1227; 22.1	479; 39	295; 17.7	265; 90.4	8469; 19.6	6779; 80.0	223; 21.8	165; 74.4	504; 21.8	324; 65.2
Richer	1270; 22.9	606; 47.7	340; 20.4	300; 88.2	7405; 17.1	6009; 81.1	209; 20.4	176; 85.0	391; 16.9	278; 72
Richest	825; 14.9	444; 53.8	333; 20	306; 92.2	5974; 13.8	4839; 81.0	109; 10.6	89; 81.7	444; 19.2	344; 78
Place of residence		***								***
Urban	1350; 24.3	675; 50.0	569; 34.2	503; 88.6	8808; 20.4	6782; 77.0	585; 57.1	564; 79.3	1041; 45	695; 67.7
Rural	4205; 75.7	1689; 40.0	1097; 65.8	964; 88.0	34,265; 79.6	26,439; 77.2	440; 42.9	347; 79.6	1273; 55	718; 57.7
Gender of the child						*				
Male	2849; 51.3	1219; 42.8	844; 50.7	739; 87.7	22359; 51.9	17351; 77.6	585;57.1	445; 78.3	1177; 50.9	738; 63.7
Female	2706; 48.7	1145; 42.3	822; 49.3	728; 88.8	20714; 48.1	15,870; 76.6	440; 42.9	366; 80.8	1137; 49.1	675; 60.6
Number of children (M; SD)	(3.77; 2.34)	(3.74; 2.40)	(2.15; 1.25)	(2.14; 1.23)	(2.09; 1.22)	(2.03; 1.16) ***	2.14; 1.40	(2.05; 1.36) ***	(3.12; 1.94)	(2.92; 1.77) ***
Mother’s age at first birth		***				***		*		***
≤20 years	3771; 67.9	1529; 40.5	1319; 79.2	1156; 87.8	19,120; 44.4	14,335; 75.0	674; 65.8	517; 77.2	1095; 47.3	610; 56.9
>20 years	1784; 32.1	835; 46.8	347; 20.8	311; 89.9	23,953; 55.6	18,886; 78.8	351; 34.2	294; 83.8	1219; 52.7	803; 66.9
Total	5555; 100	2364; 42.6	1666; 100	1467; 88.2	43,073; 100	33,221; 77.1	1025;100	811; 79.4	2314; 100	1413; 62.2

Note: N = Number; M = Mean; SD = Standard Deviation; * *p* ≤ 0.05, *** *p* ≤ 0.001.

**Table 2 ijerph-20-01755-t002:** Results from logistic regression analysis of full immunisation in South Asian countries.

	Afghanistan		Bangladesh		India		Nepal		Pakistan	
Variables	OR	95% CI	OR	95% CI	OR	95% CI	OR	95% CI	OR	95% CI
Mother’s Education										
None (ref.)	1.00		1.00		1.00		1.00		1.00	
Primary	1.79 ***	1.44–2.21	1.46	0.84–2.54	1.18 ***	1.09–1.27	1.41	0.91–2.17	2.72 ***	2.03–3.63
Secondary	2.28 ***	1.82–2.87	2.73 ***	1.53–4.84	1.35 ***	1.27–1.44	1.68 *	1.11–2.56	3.31 ***	2.54–4.32
Higher	1.85 **	1.22–2.81	2.93 **	1.43–6.00	1.29 ***	1.18–1.42	3.75 ***	1.82–7.71	4.23 ***	2.98–6.01
Wealth										
Poorest (ref.)	1.00		1.00		1.00		1.00			
Poorer	1.00	0.84–1.19	0.93	0.60–1.43	1.20 ***	1.13–1.28	1.04	0.66–1.64	1.55 **	1.18–2.03
Middle	1.04	0.87–1.23	1.44	0.87–2.38	1.46 ***	1.36–1.57	0.76	0.49–1.18	1.74 ***	1.30–2.34
Richer	1.43 ***	1.20–1.70	1.05	0.64–1.70	1.58 ***	1.45–1.71	1.30	0.78–2.17	1.97 ***	1.39–2.78
Richest	1.61 ***	1.26–2.04	1.53	0.85–2.75	1.59 ***	1.44–1.75	0.76	0.40–1.42	1.93 ***	1.31–2.83
Place of residence										
Urban (ref.)	1.00		1.00		1.00		1.00		1.00	
Rural	0.98	0.83–1.16	1.10	0.77–1.58	1.30 ***	1.22–1.39	1.14	0.83–1.58	1.11	0.89–1.37
Gender of the child										
Male (ref.)	1.00		1.00		1.00		1.00		1.00	
Female	0.99	0.88–1.10	1.11	0.82–1.51	0.95 *	0.90–0.99	1.11	0.81–1.52	0.86	0.71–1.03
Number of children	1.02 *	1.00–1.04	1.07	0.94–1.21	0.91 ***	0.89–0.93	0.91	0.81–1.02	0.97	0.92–1.02
Mother’s age at first birth										
Age ≤ 20 years	1.00		1.00		1.00		1.00		1.00	
Age > 20 years	1.34 ***	1.19–1.50	1.01	0.67–1.54	1.11 ***	1.05–1.16	1.29	0.90–1.84	1.01	0.83–1.23
Overall Model X^2^	185.42 ***		31.96 ***		748.04 ***		41.76 ***		298.86 ***	
Cox & Snell R^2^	0.03		0.019		0.017		0.40		0.123	
−2 LL	7391.86		1174.17		45575.68		995.93		2714.33	

Note: * *p* ≤ 0.05. ** *p* ≤ 0.01. *** *p* ≤ 0.001; 2 LL = Log likelihood; OR = Odd Ratio; CF = Confidence Interval.

## Data Availability

Available on request from corresponding.

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
