# Peer review of "Inequalities in Childhood Immunisation in South Asia"

_ijerph, 2023, doi:10.3390/ijerph20031755_

Round 1
Reviewer 1 Report
- Line 15: Check spelling “useing”
- Line 25: Please revise “occlusion” to “conclusion”.
- Lines 62-64: The WHO fully immunized child rate is evaluated at exactly 12 months, whereas your study was implemented in children with higher ages. This should be discussed in the manuscript.
- The Introduction section is interesting, but it is quite lengthy. In order to keep it more focused, I would suggest to move lines 68-94 to the Discussion section.
- Line 80: Lack of knowledge about vaccines has indeed been associated with decreased vaccine utilization. However this can and should be easily addressed, an interesting publication has shown the role of engaging general practitioners as well as pediatricians in discussion with parents regarding vaccines, see PMID 35335036.
- The Methods section should start by briefly explaining the methodology of the DHS. From the current description, it is hard to understand by an outside reader exactly how it was performed, whether or not the methodology was similar in all countries included in the study, what “the seventh phase” represents, etc.
- Also, it should be stated how often the DHS is repeated in order to understand why those years were chosen for each country. Because at a first impression, it is unclear why one would consider data from 2015 in a country to be comparable and analyzed in a pooled manner with data from a different country in 2020-2021.
- Lines 113-115: This description is a bit unclear: If the children had ages between 12 and 23 months at the time of evaluation, then why add the extra filter of being born in the last five years? They had been born between 12 and 23 months previously.
- Lines 121-123: This is a repetition of information already presented in the introduction.
- Line 128-132: It is not clear why this reassignment was needed. These phrases seem to repeat the information on the previous lines.
- Line 136: What do you mean by: “limited”?
- In section 2.1.2. lines 151-153 there is no need to specify the exact codes that were used, because it makes no difference to the reader whether you used 0/1 or 1/2 to code for gender.
- Line 159: independent samples t-test is only appropriate for parametrically distributed variables. This also applies to reporting medians and SD. Was variable distribution checked?
- Line 187: The term “correlated” should only be used for correlation analysis between two continuous variables. Otherwise, the term “association” should be used instead.
- All abbreviations should be defined in a table legend, including those for statistical tests (LL)
- The discussion should also address the very important topic of the anti-vaccine movement (see PMID 35096665) and to which extent it is an important factor in each of the surveyed countries.
- Line 305: “The findings are generalisable” this is an overstatement and should be nuanced in the text. No study is generalizable to everyone.
Author Response
- Line 15: Check spelling “useing”
 Thank you. We corrected this.
- Line 25: Please revise “occlusion” to “conclusion”.
 We corrected this.
- Lines 62-64: The WHO fully immunized child rate is evaluated at exactly 12 months, whereas your study was implemented in children with higher ages. This should be discussed in the manuscript.
 As per the WHO recommendation, a child was considered to be fully immunised if s/he received a BCG vaccine at birth or soon after; three doses each of DPT and polio vaccine at 6, 10 and 14 weeks of age; and the measles vaccine at 9 months or soon thereafter. Therefore, we selected the age group of 12 to 23 months to measure whether or not children received full vaccination, and the selection of the dependent variable is based on previous studies.
- The Introduction section is interesting, but it is quite lengthy. In order to keep it more focused, I would suggest to move lines 68-94 to the Discussion section.
ïƒ Thank you for the suggestion. We also considered moving lines 68 to 94 to the discussion sections as per your suggestions. However, as we re-read the paper, we again thought that it is also essential to introduce the context of conducting the research in the global context as well as South Asian content. We decided to keep these two paragraphs in the introduction section.
- Line 80: Lack of knowledge about vaccines has indeed been associated with decreased vaccine utilization. However this can and should be easily addressed, an interesting publication has shown the role of engaging general practitioners as well as pediatricians in discussion with parents regarding vaccines, see PMID 35335036.
ïƒ Thank you for the suggestion for the article. This article provides interesting information on "vaccine hesitancy," and we added this article as one of our references.
- The Methods section should start by briefly explaining the methodology of the DHS. From the current description, it is hard to understand by an outside reader exactly how it was performed, whether or not the methodology was similar in all countries included in the study, what “the seventh phase” represents, etc.
ïƒ Thank you for the suggestions. We also added a brief paragraph introducing the methodology of the DHS.
- Also, it should be stated how often the DHS is repeated in order to understand why those years were chosen for each country. Because at a first impression, it is unclear why one would consider data from 2015 in a country to be comparable and analyzed in a pooled manner with data from a different country in 2020-2021.
ïƒ Thank you. We clarified it in the text.
- Lines 113-115: This description is a bit unclear: If the children had ages between 12 and 23 months at the time of evaluation, then why add the extra filter of being born in the last five years? They had been born between 12 and 23 months previously.
ïƒ Thank you for the comment. We added the explanation of it in the manuscript.
- Lines 121-123: This is a repetition of information already presented in the introduction.
ïƒ Thank you. We deleted it.
- Line 128-132: It is not clear why this reassignment was needed. These phrases seem to repeat the information on the previous lines.
 Thank you. We reorganized these phrases.
- Line 136: What do you mean by: “limited”?
 Thank you. We corrected it.
- In section 2.1.2. lines 151-153 there is no need to specify the exact codes that were used, because it makes no difference to the reader whether you used 0/1 or 1/2 to code for gender.
 Thank you. We corrected it.
- Line 159: independent samples t-test is only appropriate for parametrically distributed variables. This also applies to reporting medians and SD. Was variable distribution checked?
 Thank you. We corrected it.
- Line 187: The term “correlated” should only be used for correlation analysis between two continuous variables. Otherwise, the term “association” should be used instead.
 Thank you. We corrected it.
- All abbreviations should be defined in a table legend, including those for statistical tests (LL)
 Thank you. We corrected it.
- The discussion should also address the very important topic of the anti-vaccine movement (see PMID 35096665) and to which extent it is an important factor in each of the surveyed countries.
 Thank you. We did it.
- Line 305: “The findings are generalisable” this is an overstatement and should be nuanced in the text. No study is generalizable to everyone.
 Thank you. We corrected it.
Reviewer 2 Report
This comparative analysis describes country specific nuances for full immunization in selected south Asia countries with available secondary data in an attempt for identifying differentials. Followings suggestions are for consideration
1. The rural urban differences as reflected in India is impact of the Routine Immunisation (RI) programme of India is focused Rural areas under National Rural Health Mission (NRHM) in the country. The National Urban Health Mission (NUHM) as a sub-mission of National Health Mission (NHM) has been approved by the Cabinet on 1st May 2013. NRHM strengthened the both supply and demand sides which is still to reach the same magnitude in the urban areas, in India.
2. Millennium Development Goals (MDGs) were the eight international development goals for the year 2015. SDGs are successor of MDGs. Immunization directly impacts health (SDG3) and brings a contribution to 14 out of the 17 Sustainable Development Goals (SDGs), especially in low- and middle-income countries (LMICs). Thus the discussion could be more relevant in light of SDGs.
3. Title suggestion "Determinants of Full Immunisation in South Asia"
4. The first portion of the paper requires language editing
Author Response
This comparative analysis describes country specific nuances for full immunization in selected south Asia countries with available secondary data in an attempt for identifying differentials. Followings suggestions are for consideration
- The rural urban differences as reflected in India is impact of the Routine Immunisation (RI) programme of India is focused Rural areas under National Rural Health Mission (NRHM) in the country. The National Urban Health Mission (NUHM) as a sub-mission of National Health Mission (NHM) has been approved by the Cabinet on 1st May 2013. NRHM strengthened the both supply and demand sides which is still to reach the same magnitude in the urban areas, in India.
 Thank you for explaining the rural-urban differences in vaccine utilization in India, which gave us insight into why there are rural-urban differences in vaccine utilization in India. We incorporated it in the discussion section. We added the following statement in the manuscript”
“In India, the rural-urban differences in childhood immumisation were possibly reflected due to the impact of India's Routine Immunisation (RI) programme, which focused on rural areas under the National Rural Health Mission (NRHM).”
- Millennium Development Goals (MDGs) were the eight international development goals for the year 2015. SDGs are successor of MDGs. Immunization directly impacts health (SDG3) and brings a contribution to 14 out of the 17 Sustainable Development Goals (SDGs), especially in low- and middle-income countries (LMICs). Thus the discussion could be more relevant in light of SDGs.
 Thank you for the suggestion. The study is more concerned with the implementation of SDGs.
- Title suggestion "Determinants of Full Immunisation in South Asia"
 Thank you for the suggestion. We are more inclined to present the vaccine utilization differences among children in South Asian countries. All the co-authors believe we need to focus on “inequalities” rather than “determinants.” Nevertheless, we thank you for your sincere suggestions to make this manuscript of high quality.
- The first portion of the paper requires language editing.
Thank you. We tried our best to edit the language. Furthermore, the manuscript is edited by a native speaker of the English language.
Reviewer 3 Report
This is an important study about childhood immunisations and their determinants in South Asia. I have some minor comments:
1. The representativity of the survey to the whole population of children in the countries participating is key to the robustness of the findings of the study. The authors state that the study "used nationally representative sample sizes" of this survey. Sample size has very little to do with representativity, so I don't understand what the authors want to say here? It must be awfully difficult to create a nationally representative sample in a nation as vast as India (as you correctly discuss), but also in a country at war as Afghanistan and a country with the geographies of Nepal and Afghanistan. So much more information is needed here about the details of the sampling procedures and how the representativity of the sample was cross-checked with other data sets.
2. The logistic regression used vaccination as the outcome. Many researchers would probably have preferred to "not vaccinated" as the outcome, since that would have produced odds ratios that could have been to interpret, more similar to relative risks.
3. Using overlapping socioeconomic indicators (SES) is often not a good idea, since they tend to attenuate each other and thus underestimate the true socioeconomic differences. Most social epidemiologists in such cases present at least two models, one with each SES variable analysed separately and one with all together. I suggest such a model is added to the analytic framework of this study.
4. Analysing maternal age as a continuous variable is usually not a good idea, since the association with the outcomes usually has a different shape. Why not try Teenage vs 0>20 years instead? Or a three category variable
5. The text in the results section is not very helpful for the reader. It could easily be improved by pointing to similaraties and differences between the countries, rather than just presenting each country separately.
6. There are some odd English expressions in the manuscript. On page 3 under 2.1.2. "Moreover, we limited the independent variables that impede".... Why limit such variables? I think another expression is needed here.
Author Response
- The representativity of the survey to the whole population of children in the countries participating is key to the robustness of the findings of the study. The authors state that the study "used nationally representative sample sizes" of this survey. Sample size has very little to do with representativity, so I don't understand what the authors want to say here? It must be awfully difficult to create a nationally representative sample in a nation as vast as India (as you correctly discuss), but also in a country at war as Afghanistan and a country with the geographies of Nepal and Afghanistan. So much more information is needed here about the details of the sampling procedures and how the representativity of the sample was cross-checked with other data sets.
 Thank you for the suggestion. We corrected and reorganized “this” (i.e., representativeness of the sample) throughout the manuscript.
- The logistic regression used vaccination as the outcome. Many researchers would probably have preferred to "not vaccinated" as the outcome, since that would have produced odds ratios that could have been to interpret, more similar to relative risks.
 Thank you for your opinion on whether we could also understand and interpret the “outcome variable” from a different domain.
- Using overlapping socioeconomic indicators (SES) is often not a good idea, since they tend to attenuate each other and thus underestimate the true socioeconomic differences. Most social epidemiologists in such cases present at least two models, one with each SES variable analysed separately and one with all together. I suggest such a model is added to the analytic framework of this study.
-  Thank you for your comment. We want to explore inequalities in childhood immunisation in South Asia. And we want to examine to what extent the social determinants factor (i.e., education, wealth, place of residence) affect the likelihood of childhood immunisation in the South Asian context.
- Analysing maternal age as a continuous variable is usually not a good idea, since the association with the outcomes usually has a different shape. Why not try Teenage vs 0>20 years instead? Or a three category variable
 Thank you for the suggestion. We recoded maternal age variable into two categories as Teenage (<20 years) vs 0>20 years in the analyses.
- The text in the results section is not very helpful for the reader. It could easily be improved by pointing to similaraties and differences between the countries, rather than just presenting each country separately.
 Thank you. We corrected it accordingly.
- There are some odd English expressions in the manuscript. On page 3 under 2.1.2. "Moreover, we limited the independent variables that impede".... Why limit such variables? I think another expression is needed here.
 Thank you. We corrected it accordingly.